# Rationing nursing care and organizational factors in intensive care units

**Anna Antoszewska**(ID)\*[º], **Aleksandra Gutysz-Wojnicka**(ID)[º]

Department of Nursing, School of Public Health, Collegium Medicum, University of Warmia and Mazury in Olsztyn, Olsztyn, Poland

[º] These authors contributed equally to this work.

\* anna.laguna@gmail.com

## Abstract

### Introduction

Rationing nursing care is a term that applies to various aspects of the required patient care that are omitted or their performance is delayed.

### Aim

This study aimed to identify the extent of rationing of nursing care in intensive care units (ICUs) in different types of hospitals and determine the relationship between rationing of nursing care and hospital and staff characteristics.

### Methods

This quantitative, cross-sectional, multicenter study was performed. The sample comprised 226 nurses working in ICUs in a North-East part of Poland. The Polish version of the PRINCA questionnaire methods was applied. The survey was conducted between 15 January and 31 May 2023.

### Results

There were statistically significant differences between rationing of nursing care in university/provincial hospitals and district hospitals t = 6.92 p<0.001. In provincial and university hospitals, nursing care is often omitted, leading to a lower perceived quality of nursing care (t = -3.0 p = 0.003). This is further compounded by the fact that nursing care is more likely to be rationed in units with a larger number of beds. The level of rationing of nursing care was significantly correlated with the perceived work quality and job satisfaction in both types of hospitals. The most frequently omitted aspects of nursing care included providing emotional support (university/provincial 1.27 vs. district 0.89), patient and family education (1.11 vs. 0.74), communication with external entities (1.11 vs. 0.84), and observing safe patient-handling practices (1.01 vs. 0.99).

**Data Availability Statement:** All relevant data are within the manuscript and its Supporting Information files.

**Funding:** The author(s) received no specific funding for this work.

**Competing interests:** Enter: The authors have declared that no competing interests exist.

## Conclusion

The type of hospital and organizational factors influence the rationing of nursing care. Improvements in working conditions can improve nursing care quality in ICUs.

## Introduction

Missed nursing care is defined as standard and required nursing care that is not completed or is seriously delayed. The Missed Nursing Care Model described by Kalisch and Lee (2010) [1] consists of the interaction of contributing variables such as organizational factors, hospital, unit and staff characteristics and teamwork to missed nursing, and the patient outcomes and staff satisfaction of missed nursing care.

According to Kalisch et al. [2], the rationing of nursing care applies to aspects of required patient care that are omitted (in part or in whole) or delayed. Omitted nursing care constitutes an error of omission [3]. Rationing of nursing care usually occurs when available resources are insufficient to provide appropriate quality nursing care to patients. Rationing of nursing care negatively impacts treatment outcomes, with an increase in falls, hospital-acquired infections, bedsores, and low-quality care [4–7]. Rationing nursing care has a direct effect on patients' clinical outcomes.

According to a systematic review conducted by Recio-Saucedo et al. [8], the lack of nursing care has a significant impact on patient outcomes. The review found that the absence of nursing care can lead to bedsores, treatment errors, hospital-acquired infections, patient falls, critical incidents, and rehospitalization within 30 days. In addition, the review analyzed nine studies that reported mortality rates due to the lack of care. Furthermore, four studies conducted in hospital settings revealed that the absence of care significantly reduces patient satisfaction.

Schubert et al. [9] showed that even a low level of care rationing was associated with deteriorating patient outcomes. Rationing of nursing care poses a significant threat to patient security and healthcare quality [10]. On the other hand, research has shown that increasing the number of nurses reduces the lack of care [11]. According to some predictions, a lower level of nursing care omissions will improve nursing oversight and patient treatment outcomes, such as the number of falls and the occurrence of bedsores and pneumonia [12]. Rationing of nursing care is mainly caused by insufficient nursing staff, adverse work environments, inadequate financial resources, and nurses' attitudes and knowledge [13, 14]. A significant factor contributing to the rationing of nursing care is the workload burden nurses face. High workloads increase nursing staff stress levels and limit their ability to provide comprehensive care, leading to increased instances of care omission or delay [15, 16].

In Poland, the Regulation of the Minister of Health of 16 December 2016 on the organizational standard of health care in the field of anesthesiology and intensive care regulates that in ICUs, for patients with the highest level of care, the ratio of the number of nurses per shift to the number of patients should be 1:1. If a lower level of care is needed, the number of nurses in relation to the number of patients should be 1:1.5 (2:3), or 1: 4 depending on the patient medical condition. Additionally, nurses in ICUs are required to have completed a specialization or qualification course in the field of anesthesia and intensive care nursing [17]. The number of patients treated in ICUs is constantly growing. This increase is caused by demographic changes in society, aging, and technological progress extending the average lifespan [18, 19].

The work of ICU nurses requires vast knowledge from many branches of medicine and the ability to cope with high stress levels.

Numerous studies have focused on the rationing of nursing care among general nurses; however, only one study in Poland has explored this issue among ICU nurses [20]. Identifying the organizational and staff -related factors contributing to the omission of nursing care in ICUs remains a significant research gap. Several authors have reported that various factors, such as nursing staff fatigue [20], communication and teamwork [21], coping with time pressure [22], work environments and nurse staffing [23, 24], type of hospital [25], and different roles within nursing [26], influence the rationing of nursing care. However, there are still many factors related to hospital, unit, and staff characteristics that need to be further analyzed. In 2010, Kalisch and Lee emphasized the importance of these factors on the rationing of nursing care, patient outcomes, and job satisfaction.

Therefore, to address this gap, this study aimed to investigate the relationship between different hospital types, unit characteristics, sociodemographic variables and the rationing of nursing care.

## Aim

This study aimed to identify the extent of rationing of nursing care in ICUs in different types of hospitals and determine the relationship between nursing care rationing, unit characteristics (number of beds) and staff characteristics (sociodemographic variables, job satisfaction and perceived quality of care).

## Methods

### Study design

This quantitative, cross-sectional, multicenter study was performed on 274 nurses working in intensive care units (ICU) in hospitals of various reference levels in the Warmia and Mazury Region of Poland. The convenience sampling method was used to collect the data. The data for this study were collected between 15 January and 31 May 2023. A validated research tool, such as the PRINCA questionnaire was applied. The STROBE checklist for observational cross-sectional studies was followed to report the research study [26].

### Materials and methods

This study employed the Perceived Implicit Rationing of Nursing Care (PRINCA) questionnaire and a questionnaire developed by the author to gather demographic data. The PRINCA questionnaire was originally developed by Jones in the USA in 2014 [27]. It consists of 31 questions regarding six areas of a nurse's activities: nursing care, implementation of a defined treatment plan, emotional support and education, supervision/vigilance, coordination of care and planning, and documentation of the conducted activities. Additionally, it contained two questions about patient care quality and job satisfaction. The respondents were required to indicate how often during the seven latest duties/shifts they could not complete each of the 31 described nursing activities because of a lack of resources (personnel or time). The frequency of each of the 31 actions was rated according to the following scale: never = 0, rarely = 1, sometimes = 2, and often = 3. A "not applicable" option was applied to patients who required no specific nursing action during the seven shifts. The total score was the mean number of points for the questions for which one of the above options was selected (i.e. "not applicable" responses were excluded). Therefore, the result ranged between 0–3, and it could be interpreted as follows: higher scores mean more frequent rationing of nursing care. The answers to the two questions

concerning patient care quality and general job satisfaction lie within the range of 0–10, where higher numbers denote better opinions on care quality and higher job satisfaction. The PRINCA questionnaire has been validated in many countries [27–30]. With the original author's consent, this study employed a Polish version of the PRINCA questionnaire culturally adapted by Uchmanowicz et al. [30]. Cronbach's alpha for the Polish version of the questionnaire was 0.98, which denotes a very high level of reliability and is close to the value provided by the original questionnaire [27].

Based on the literature review, the current study analyzed quantitative variables related to unit characteristics such as a nurse-patient ratio, number of beds in the ICU, variables related to staff characteristics such as sex, age, level of education, number of working hours per month, working system, job satisfaction and perceived quality of care to assess their potential impact on the rationing of nursing care in different hospital types.

## Participants

The participants included female and male nurses working in 13 adult ICUs in the Warmia and Mazury region of Poland. According to a list provided by the Voivodeship Office in Olsztyn [31], 19 ICUs were in this region of Poland at the time of the study. Consent to participate in the study was obtained from 13 hospitals (68.4% of all ICUs in the region). Three withheld permission from the remaining six hospitals, and the other three were not invited to participate due to organizational problems. Pediatric ICUs and neonatal ICUs were excluded. In total, 396 nurses were working in the adult ICUs in the region. The sample size was determined using an arithmetic mean estimation of the PRINCE questionnaire. The study's inclusion criteria for nurses were employment at an adult ICU for at least six months as a nurse, providing direct patient care at a public or non-public hospital, and providing informed consent for participating in the study. Exclusion criteria included being employed as an anesthesia and intensive care nurse at a hospital unit other than the adult ICU, performing functions other than direct patient care at an ICU (e.g. head nurse, nursing coordinator), and not providing informed consent for participation in the study.

## Data gathering and statistical procedures

Managers of healthcare facilities invited to the study provided consent before the research. Questionnaires and the informed consent form were administered individually to all nurses who met an inclusion criterion in an envelope by one of the investigators. Of the 274 potential respondents in the 13 ICUs (69.2% of all anesthesia and intensive care nurses in the region), 260 nurses (94.9%) met the inclusion criteria and received the respondent package. The 34 (13.07%) questionnaires that were not filled in or were incomplete were excluded from the analysis. This decision was made to ensure the reliability of the results by avoiding imputation techniques that could introduce additional errors. Only fully completed questionnaires were further analyzed, allowing the study to maintain data quality and obtain reliable statistical conclusions despite potential limitations related to the sample's representativeness. To further minimize potential bias and ensure the reliability of the results, the participant selection process was carefully supervised, and data collection methods were standardized and uniform for all respondents, regardless of their place of work.

The data were entered into a Microsoft Excel spreadsheet database, and the results were analyzed using TIBC Statistica 13.3. The two groups of hospital type were identified (district hospital– 1st degree of reference and provincial and university hospitals– 2nd level of reference). The Pearson chi-square test ($\chi 2$) was used to analyze the distribution of sociodemographic variables in two groups of hospital type. Quantitative variables (expressed as figures)

were analyzed by calculating the mean, median, minimum, maximum, and standard deviation. Quantitative variables in the two hospital-type groups were compared using variance ANOVA (F) analysis. Descriptive statistics were used to describe the sample characteristics. Student's t-test was used to compare the significance of the mean results for the PRINCA question distribution in two hospital types. Correlations between quantitative variables were analyzed with Pearson's (r) correlation coefficients. The statistical significance level was set at $p < .05$.

The relationship strength was interpreted according to the following pattern: $r < 0.2$ = weak relationship, $0.2 < r < 0.4$ = distinct, but low and moderate relationship when $0.4 < r < 0.7$, considerable when $0.7 < r < 0.9$, and very strong when $r > 0.9$.

An attempt was made to identify significant predictors of rationing of nursing care (PRINCE) from the analyzed variables. For this purpose, a stepwise multiple linear regression analysis was performed in both types of hospitals. Determination coefficients were also calculated using the formula: $R^2 = r^2 \times 100\%$ ($R^2$—coefficient of determination; r–Pearson's correlation coefficient).

## Ethical procedures

The respondents were informed before the study that participation was voluntary and that anonymity would be maintained throughout the study. Permission to conduct the study was obtained from each hospital manager before the study. The full anonymity of the study participants was maintained. The study protocol was approved by the Committee for Research Ethics of the University of Warmia and Mazury in Olsztyn (DECISION no. 16/2022). Each participant received written information regarding the study's objectives and an informed consent form. Participation was optional and anonymous. The informed consent form contained information that participants were free to discontinue the survey whenever they wished, without facing any repercussions or needing to justify their decision. To ensure anonymity, the research instruments were devoid of any identifying features. Submitting the filled-out questionnaire and signing the informed consent form was interpreted as consent to participate in the research.

## Results

### Sample characteristics

This study included 226 respondents from 13 ICUs in the Warmia and Mazury region of Poland. Out of the total, 128 participants worked in district hospital ICUs and 98 in provincial and university hospitals. There was a statistically significant difference in the distribution of sociodemographic variables such as age, level of education, working system, nurse-patient ratio, and number of ICU beds in the two hospital groups. The ICU nurses in university/provincial hospitals were younger, more often had a master's degree in nursing, worked more often on 12 hours day/night shifts, took care of a smaller number of patients during the working shift and worked in larger ICUs than the nurses in district hospitals.

The detailed demographic characteristics of the groups of participants are shown in Table 1.

### Care rationing level

In provincial and university hospitals, the mean score of responses to questions related to missed nursing care was 0.84 (0–3 scale), 7.7 (0–10 scale) for care quality, and 7.01 (0–10 scale) for job satisfaction. In district hospitals, the mean scores for rationing of nursing care were 0.38, 8.3 for care quality, and 6.84 for job satisfaction. This indicates that nursing care is

**Table 1. Distribution of sociodemographic variables in two groups of hospital type.**

| Feature | | University/Provincial hospitals | District hospitals | Overall | χ2 | p |
|---|---|---|---|---|---|---|
| | | N (%) | N(%) | N (%) | | |
| Sex | Female | 87 (88.78) | 116 (90.63) | 203 (89.82) | 0.21 | 0.64 |
| | Male | 11 (11.22) | 12 (9.38) | 23 10.18 | | |
| Age (years) | 20–30 | 19 (19.39) | 22 (17.19) | 41 (18.14) | 17.44 | <0.001 |
| | 31–40 | 39 (39.80) | 22 (17.19) | 61 (26.99) | | |
| | 41–50 | 24 (24.49) | 45 (35.16) | 69 (30.53) | | |
| | 51 and above | 16 (16.33) | 39 (30.47) | 55 (24.34) | | |
| Education | Medical/nursing secondary | 10 (10.20) | 30 (23.44) | 40 (17.70) | 11.37 | 0.003 |
| | Bachelor of nursing studies | 24 (24.49) | 41 (32.03) | 65 (28.76) | | |
| | Master's degree in nursing | 64 (65.31) | 57 (44.53) | 121 (53.54) | | |
| Number of hours spent at work per month | below 100 | 1 (1.02) | 6 (4.69) | 7 (3.10) | 1.65 | 0.19 |
| | 100–180 | 53 (54.08) | 69 (53.91) | 122 (53.98) | | |
| | 181–230 | 31 (31.63) | 37 (28.91) | 68 (30.09) | | |
| | 231–300 | 12 (12.24) | 14 (10.94) | 26 (11.50) | | |
| | More than 300 | 1 (1.02) | 2 (1.56) | 3 (1.33) | | |
| Work system | Single shift/ 8 hours day shift | 1 (1.02) | 16 (12.50) | 17 (7.52) | 13.04 | <0.001 |
| | 12 hours/ day/night shifts | 97 (98.98) | 112 (87.50) | 209 (92.48) | | |
| Number of patients under care (Nurse-Patient ratio) | One | 0 (0.00) | 2 (1.56) | 2 (0.88) | 62.75 | <0.001 |
| | Two | 96 (97.96) | 75 (58.59) | 171 (75.66) | | |
| | Three | 0 (0.00) | 37 (28.91) | 37 (16.37) | | |
| | Four | 2 (2.04) | 14 (10.94) | 16 (7.08) | | |
| Number of beds in the ward | 3–7 | 3 (3.06) | 113 (88.28) | 116 (51.33) | 193.82 | <0.001 |
| | 8–20 | 95 (94.94) | 15 (11.72) | 110 (48.67) | | |

"rarely" rationed in both types of hospitals; however, there were statistically significant differences between the rationing of nursing care in university/provincial hospitals and district hospitals t = 6.92 p<0.001. Nursing care is omitted more often in provincial/university hospitals. Moreover, the perceived quality of nursing care is lower in provincial/university hospitals (t = -3.0 p = 0.003). The mean results of the PRINCA questionnaire are presented in Table 2.

## Factors affecting care rationing level

The rationing of nursing care was significantly correlated with the perceived work quality and job satisfaction in both provincial/university and district hospitals. Higher levels of nursing care rationing are correlated with lower subjective ratings of nursing care quality and job satisfaction. The correlation data are presented in Table 4.

**Table 2. PRINCA average scores.**

| PRINCE | Provincial/university hospitals | | District hospitals | | t | p |
|---|---|---|---|---|---|---|
| | N | Mean (SD), -95%– 95%, Me, Min.–Max. | N | Mean (SD), -95%– 95% Me, Min.–Max., | | |
| Care Rationing | 98 | 0.84 (0.59), 0.72–0.96 0,74, 0–2.29 | 128 | 0.38 (0.42), 0.30–0.45 0.26, 0–2.32 | 6.92 | <0.001 |
| Quality of care | 98 | 7.70 (1.45), 7.41–8.00 8.0, 4.0–10.0 | 128 | 8.30 (1.52), 8.04-.8.57 9.4–10.0 | -3.00 | 0.003 |
| Job Satisfaction | 98 | 7.01 (1.73), 6.66–7.36 7.0, 3.0–10.0 | 128 | 6.84 (1.94), 6.50-.7.18 7.3–10 | 0.70 | 0.48 |

**Table 3. Rationing of nursing care in two different hospital types and demographic variables.**

| Feature | Provincial/university hospitals | | | District hospitals | | |
|---|---|---|---|---|---|---|
| | PRINCA | | | PRINCA | | |
| | Care rationing Mean(SD) | Quality of care Mean(SD) | Job satisfaction Mean(SD) | Care rationing Mean(SD) | Quality of care Mean(SD) | Job satisfaction Mean(SD) |
| Sex | F = 0.61 p = 0.43 | F = 2,23 p = 0.12 | F = 0.02 p = 0.87 | F = 3.52 p = 0.06 | F = 2.16 p = 0.14 | F = 4.89. p < 0.02 |
| Female | 0.86(0.61) | 7.78(1.39) | 7.00(1.75) | 0.40(0.43) | 8.24(1.56) | 6.72(1.91) |
| Male | 0.71(0.41) | 7.09(1.81) | 7.09(1.64) | 0.16(0.21) | 8.92(0.90) | 8.00(1.91) |
| Age (years) | F = 1.70 p = 0.17 | F = 0.73 p = 0.53 | F = 1.37 p = 0.25 | F = 1.73 p = 0.16 | F = 2.88. p < 0.04 | F = 2.48 p = 0.06 |
| 20–30 | 0.67(0.47) | 7.89(0.99) | 7.68(1.49) | 0.28(0.30) | 8.41(1.44) | 7.64(1.84) |
| 31–40 | 0.98(0.61) | 7.69(1.47) | 6.77(1.83) | 0.37(0.45) | 8.14(1.73) | 7.27(1.70) |
| 41–50 | 0.72(0.62) | 7.88(1.75) | 6.79(1.74) | 0.33(0.37) | 8.76(1.28) | 6.53(2.06) |
| 51 and above | 0.89(0.58) | 7.25(1.39) | 7.13(1.67) | 0.50(0.49) | 7.82(1.59) | 6.49(1.88) |
| Number of beds in the ward | F = 1.06 p = 0.30 | F = 1.38 p = 0.24 | F = 1.82 p = 0.18 | F = 0.02 p = 0.87 | F = 0.96 p = 0.32 | F = 6.004 p = 0.94 |
| 3–7 | 0.49(0.62) | 8.67(1.53) | 8.33(0.58) | 0.37(0.42) | 8.26(1.58) | 6.83(1.95) |
| 8–20 | 0.85(0.59) | 7.97(1.45) | 6.97(1.74) | 0.39(0.37) | 8.67(1.18) | 6.87(196) |

No statistically significant difference was found between the level of rationing of nursing care, sense of quality of care and job satisfaction in intensive care units and demographic variables such as gender, age, and ward size in the two hospital groups. Detailed data are presented in Table 3.

The rationing of nursing care was significantly negatively correlated with quality of care and job satisfaction in both provincial/university hospitals and district hospitals. Thus, higher rationing of nursing care was associated with lower subjective ratings of nursing care quality and job satisfaction. The correlation data are presented in Table 4.

Statistically significant positive correlations were also found between the sense of quality of nursing care and job satisfaction. When healthcare providers offered better quality nursing care, employees' job satisfaction increased proportionally.

Since both quality of care and job satisfaction have a significant impact on rationing nursing care level and considering the significant relationship between quality of care and job satisfaction, attempts were made to select significant predictors from the group of analyzed variables and sociodemographic variables. For this purpose, a stepwise multiple linear regression analysis was performed in both types of hospitals. Coefficients of determination were also calculated according to the formula: $R^2 = r^2 \times 100\%$ ($R^2$—coefficient of determination; r—Pearson's correlation coefficient). As a result of the analysis, the model describing omitted nursing care included quality of care, with 16% determining the results for provincial/university hospitals and 12% determining results for district hospitals. The job satisfaction variable was eliminated when significant rationing nursing care predictors were identified.

**Table 4. Correlations between rationing of nursing care and quality of care and job satisfaction.**

| Correlations r-Person | Provincial/University hospitals | | | District hospitals | | |
|---|---|---|---|---|---|---|
| | PRINCA | Quality of care | Job satisfaction | PRINCA | Quality of care | Job satisfaction |
| PRINCA | 1 | | | 1 | | |
| Quality of care | -0.40 p<0.001 | 1 | | -0.34 p<0.001 | 1 | |
| Job satisfaction | -0.22 p<0.02 | 0.35 p<0.001 | 1 | -0.28 p<0.02 | 0.46 p<0.001 | 1 |

**Table 5. Multiple linear regression analysis for predictors of nursing care rationing.**

| PRINCA predictors emerging from multiple regression | | | | |
|---|---|---|---|---|
| **Provincial/University hospitals** | **R** | **R$^2$** | **F** | **p** |
| Quality of care | -0.40 | 0.16 | 18.60 | <0.001 |
| **District hospitals** | | | | |
| Quality of care | -0.34 | 0.12 | -2.91 | 0.004 |

A significant negative relationship at the mean level was found between the quality of care and the level of nursing rationing in each hospital type. This means that a higher quality of care score moderately determines a lower level of nursing rationing (Table 5). A significant negative relationship between job satisfaction and PRINCA level was found at a low level in each hospital type. This means that a higher job satisfaction score insignificantly determines a lower level of nursing rationing.

## The most frequently rationed areas of care

The most frequently omitted aspects of nursing care in intensive care units, both in district and provincial/university hospitals, included providing emotional support (university/provincial 1.27 vs. district hospitals 0.89), patient and family education (1.11 vs. 0.74), communication with external entities (1.11 vs. 0.84), and observing safe patient-handling practices (1.01 vs.0.99). Additionally, for nurses in provincial/university, the rationing level was high for responses to questions about assisting in patient mobilization (1.17), assisting in bowel or bladder evacuation (e.g., crotch hygiene maintenance) (1.07), changing sheets (1.06), changing body position (1.02), assistance in eating (0.96), and the possibility of introducing measures promoting physical comfort (1.04). The distribution of responses to the individual questions in the PRINCA questionnaire, broken down into provincial/university and district hospitals, is shown in S1 Table. The statistical analysis revealed significant differences between the two groups of hospital type, evidenced by p-values (p < .001) in the case of each question (S1 Table).

## Discussion

This study aimed to identify the extent of rationing of nursing care in different types of hospitals and determine the relationship between the rationing of nursing care and unit (number of beds) and staff characteristics (sociodemographic variables, job satisfaction and perceived quality of care). This study found that rationing of nursing care in ICU is a greater problem in provincial/university hospitals, which serve as the second reference level hospitals, than in the first reference level district hospitals. This finding applied to all aspects of nursing care. Moreover, the quality of care perceived by ICU nurses was significantly lower in provincial/university hospitals.

Similar findings were demonstrated in studies conducted by Zeleníková et al. [32] and Bragadóttir et al. [33], which showed that nurses at university hospitals reported more missed care than those in general hospitals. However, the contrasting findings were also reported by Zeleníková et al. [25], who reported that nursing care in all aspects was rationed more often in small hospitals than in large ones.

The present study showed that the characteristics of ICU nurses in both types of hospitals were significantly different. The ICU nurses in university/provincial hospitals were younger, more often had a master's degree in nursing, worked more often on 12 hours day/night shifts, took care of a smaller number of patients during a working shift and worked in larger ICUs

than the nurses in district hospitals. These findings suggest that nurse characteristics and organizational factors influence nursing care rationing. In this study, the number of beds per ICU was one of the statistically significant factors affecting the rate of nursing care omission. A significantly larger number of beds is linked to a higher workload on wards in provincial/university hospitals. Zeleníková et al. [25] also reported that the number of nurses was significantly higher in large hospitals than in medium-sized hospitals. Work in provincial/university hospitals involves participation and assistance to a larger number of highly specialist therapeutic and diagnostic procedures (such as dialysis, specialist examinations, and Extracorporeal Membrane Oxygenation (ECMO) due to the severity and complexity of patient medical conditions than in district hospitals.

Provincial/university hospitals also play a role in the continuous education and training of medical professionals. These additional tasks influence the physical and mental workload of ICU nurses. The study conducted in a neonatal ICU by Tubbs-Cooley et al. [34] showed that workload affected the level of omitted nursing care. Other authors [15, 16] highlight the significant relationship between the high mental and physical workload of ICU nurses and organizational factors like the patient-to-nurse ratio. It emphasizes the need to address these factors to effectively manage the workload, suggesting that both nurse and patient characteristics contribute to the workload in ICUs. However, a precise analysis of the higher levels of nursing care rationing requires further study. The findings of this study highlight the need to acknowledge the dual role of provincial and university hospitals. The findings of this study showed that sociodemographic factors had a minor impact on care rationing in ICUs. Similar conclusions were drawn in a study conducted in ICUs by Młynarska et al. [20]. Fatigue was identified as a significant cause of nursing care rationing in ICUs.

The most frequently omitted aspects of nursing care in district hospitals are actions such as education, communication and emotional support. On the other hand, nurses in provincial/university hospitals had to rationalize fulfilling patients' basic needs, such as hygiene, changing sheets, and changing patients' positions. Although the participants reported various nursing actions as missed, the mean results showed that the frequency of nursing care rationing was relatively low, from "never" to "rarely." Compared with nurses working in other hospital wards, nurses in ICUs in Poland rationed care at a "slightly" lower level. This was also demonstrated in a study conducted by Bragadóttir, Kalisch, and Tryggvadóttir [35] and Kalankova et al. [36], who reported that nursing care was omitted in ICUs significantly less frequently than in other hospital units. In Poland, other authors reported that rationing of nursing care among hospital nurses happened between "rarely" and "sometimes" [30, 37].

## Limitations

This study had some limitations. The participants were selected in a non-randomized manner, and only hospitals and nurses who provided informed consent were included. Moreover, owing to the sensitivity of the issue, many participants did not complete the questionnaires, making it necessary to exclude them from further analyses. However, the strength of this study is that the sample was collected from 13 hospitals out of 19 in this region of Poland. The group from which participants were chosen was homogeneous and included only nurses working in adult ICUs.

Although the study provides important conclusions regarding the rationing of nursing care in ICUs in the Warmia and Mazury region of Poland, the generalizability of these results to other regions or countries requires caution because of specific regional and cultural factors that may influence the organization and practice of health care. Expanding the geographical range of subsequent studies could improve the applicability of the findings across diverse

regions. In addressing the limitations of the study, it is important to acknowledge that while a quantitative approach offers a comprehensive overview of care rationing, it may not fully capture the underlying reasons or the nuanced experiences of the nursing staff. Future research could benefit from incorporating qualitative methods, such as interviews or focus groups, to gain deeper insights into the organizational factors influencing care rationing. Additionally, the study's cross-sectional design provides only a temporal glimpse, limiting the ability to observe how the rationing of nursing care evolves over time. A longitudinal study design could be instrumental in understanding the impact of organizational changes on care rationing within hospitals over extended periods.

## Conclusions

ICUs in provincial and university hospitals may differ greatly from those in district hospitals. The differences are related to organizational factors (number of beds, nurse-patient ratio, working system) and nurse-related factors (age, education, job satisfaction). It can also be presumed that the workload is much greater in provincial/university hospitals. The findings suggest that the level of rationing of nursing care may be higher in university/provincial hospitals than in district hospitals due to organizational, personnel and workload differences between different types of hospitals. Nursing managers in ICUs should regard nursing care rationing as a serious threat to the safety and quality of nursing care in the units. The impact of workload on nursing care rationing requires further research.

## Implications for nursing practice

Organizational factors and nurse-related factors may affect the extent of rationing of nursing care in ICUs. Effective monitoring of workload and appropriate staffing seems to be a key element in decreasing the rationing of nursing care. Implementing interventions that affect work organization and the effectiveness of teamwork may also contribute to improving the quality of nursing work. The negative effects of omitted patient care tasks in hospitals emphasize the importance of further research on the factors that affect the performance of nursing activities.

## Supporting information

**S1 Appendix. PRINCE questionnaire.**
(PDF)

**S1 Dataset.**
(XLSX)

**S1 Table. Distribution of responses in hospitals.**
(DOCX)

## Acknowledgments

The authors would like to thank all nurses who participated in the study.

## Author Contributions

**Conceptualization:** Anna Antoszewska, Aleksandra Gutysz-Wojnicka.

**Data curation:** Anna Antoszewska, Aleksandra Gutysz-Wojnicka.

**Formal analysis:** Anna Antoszewska, Aleksandra Gutysz-Wojnicka.

**Investigation:** Anna Antoszewska, Aleksandra Gutysz-Wojnicka.

**Methodology:** Anna Antoszewska, Aleksandra Gutysz-Wojnicka.

**Project administration:** Anna Antoszewska, Aleksandra Gutysz-Wojnicka.

**Resources:** Anna Antoszewska.

**Software:** Anna Antoszewska, Aleksandra Gutysz-Wojnicka.

**Supervision:** Aleksandra Gutysz-Wojnicka.

**Validation:** Aleksandra Gutysz-Wojnicka.

**Visualization:** Anna Antoszewska.

**Writing – original draft:** Anna Antoszewska.

**Writing – review & editing:** Aleksandra Gutysz-Wojnicka.

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
