## [Decision Letter · Decision Letter 0]

20 Feb 2024

PONE-D-24-02427The organizational dimensions of rationing nursing care in intensive care unitsPLOS ONE

Dear Dr. Antoszewska,

Thank you for submitting your manuscript to PLOS ONE. After careful consideration, we feel that it has merit but does not fully meet PLOS ONE’s publication criteria as it currently stands. Therefore, we invite you to submit a revised version of the manuscript that addresses the points raised during the review process.

We look forward to receiving your revised manuscript.

Kind regards,

Othman A. Alfuqaha, Ph.D.

Academic Editor

PLOS ONE

Additional Editor Comments:

Dear authors,

I hope this message finds you well. I am writing to provide feedback on the recent review of your manuscript submitted to our journal. The reviewer has offered valuable insights and recommendations for improving your work, and I kindly request your attention to address these suggestions in your revision.

In particular, the reviewer highlighted the importance of thorough proofreading to ensure linguistic accuracy and clarity. Additionally, it was noted that careful attention should be given to the statistical methodologies employed in your study to ensure their correctness and validity.

I highly recommend that you carefully consider these points and make the necessary revisions to enhance the quality and rigor of your manuscript. Your attention to these details will greatly contribute to the overall strength of your work and its suitability for publication in our journal.

I look forward to receiving your revised version and am available to provide any further guidance or assistance you may require throughout the revision process.

Thank you for your attention to this matter.

Best regards,

Dr. Alfuqaha

Reviewers' comments:

Reviewer's Responses to Questions

**Comments to the Author**

1. Is the manuscript technically sound, and do the data support the conclusions?

Reviewer #1: Yes

Reviewer #2: Yes

2. Has the statistical analysis been performed appropriately and rigorously? 

Reviewer #1: No

Reviewer #2: Yes

3. Have the authors made all data underlying the findings in their manuscript fully available?

Reviewer #1: Yes

Reviewer #2: Yes

4. Is the manuscript presented in an intelligible fashion and written in standard English?

Reviewer #1: Yes

Reviewer #2: Yes

5. Review Comments to the Author

Reviewer #1: Dear authors

Thank you for your efforts to do this research. Please See my comments in below.

General Comment:

The manuscript should be edited by a person who fluent in English language.

Additional Comments:

Title:

The phrase " the organizational dimensions" could be removed from the title. Additionally, it is better to add the phrase "and organizational factors"

Abstract:

Please add the study time.

line 30: If the journal guidelines allow, remove the word "discussion"

Introduction section:

In general, I did not understand why the authors addressed only the number of beds and the type of hospital among the organizational factors. Because I think other organizational factors can contribute to the rationing of nursing care. The authors have also collected data on the number of work shifts, working hours, and patients under care, which are also considered organizational factors. Therefore, I wonder why the organizational factors were limited only to the number of beds and the type of hospital. I suggest that if there is no specific logic for choosing these two types of factors, it is better to state the organizational factors instead of emphasizing these two factors. In addition, it is better to clearly define the organizational factors in the introduction section.

The introduction section is not written logically and coherently. It does not show the importance and necessity of conducting the study.

A brief review of the literature on the research topic is necessary.

I think that nursing workload in ICUs is a key factor in rationing nursing care. Therefore, I suggest that the introduction be developed using a related literature review of nursing workload in ICUs. For example, authors can use the following articles:

https://onlinelibrary.wiley.com/doi/full/10.1002/nop2.785

https://onlinelibrary.wiley.com/doi/abs/10.1111/nicc.12548

Method section:

I was surprised to see so many decimals in this manuscript. Please enter numbers to two decimal places in all sections of the manuscript.

Please explain how to estimate the sample size.

Line 80: Is the number 14 above 2014 related to the reference? Please correct it.

I strongly recommend using appropriate regression models in data analysis.

Results section:

I emphasize the data analysis must be checked by a statistical consultant.

Table 1: please merge the (N) and (%) columns. as follows: N (%).

Table 1: please delete the percent (%) from the numbers. for example, write 88.78 instead of 88.78%.

Table 2: Does the letter M mean "Mean"? Please write in full.

Table 2: Please merge "M" and "SD" columns.

Table 2: Please write the p-value of 0.00000000005 as below. P <001. AS WELL AS, please apply this comment for table 5.

This type of P-value reporting is not correct. In such a situation, the follow-up is reported as P <001. Please apply this comment in similar sections as well.

Discussion section

I suggest that the discussion section be developed using a related literature review. For example, authors can use the following articles:

https://onlinelibrary.wiley.com/doi/full/10.1002/nop2.785

https://onlinelibrary.wiley.com/doi/abs/10.1111/nicc.12548

Best Regards

Reviewer #2: The manuscript titled "The organizational dimensions of rationing nursing care in intensive care units" presents a significant contribution to the field of nursing and healthcare management by exploring the impact of hospital organizational factors on the extent of nursing care rationing in intensive care units (ICUs). The study is well-structured, with a clear aim to examine the influence of the number of beds and type of hospital on nursing care rationing, alongside exploring organizational factors affecting this issue.

One of the strengths of the study is its comprehensive methodology, utilizing the Perceived Implicit Rationing of Nursing Care (PRINCA) standardized questionnaire to collect data from a substantial sample of nurses. The findings are insightful, highlighting that a larger number of beds and voivodeship hospital settings are associated with a greater extent of nursing care rationing, pointing towards significant organizational influences on care quality.

However, there are several areas where the study could be improved:

The study is conducted in a specific region of Poland, which may limit the applicability of its findings to other regions or countries with different healthcare systems and organizational structures. Future research could benefit from a broader geographical scope to enhance the generalizability of the results. Maybe add this point to the limiatation section.

While the quantitative approach provides a solid overview of the extent of care rationing, integrating qualitative methods could offer deeper insights into the reasons behind care rationing and the subjective experiences of nursing staff. This could include interviews or focus groups with nurses to explore the nuances of organizational factors affecting care rationing.

The manuscript could be enriched by discussing potential intervention strategies or recommendations based on the findings. Identifying actionable steps that hospitals can take to mitigate the rationing of nursing care would provide practical value to hospital administrators and policymakers. Maybe you should add this into the discussion of the results.

The cross-sectional design of the study provides a snapshot of the situation but does not account for changes over time. A longitudinal study design could offer insights into how organizational changes within hospitals affect the rationing of nursing care over time.

The manuscript states that all relevant data are within the manuscript and its Supporting Information files. Ensuring the accessibility of this data for verification and future research would enhance the manuscript's transparency and credibility.

By addressing these areas, the manuscript could offer a more comprehensive view of the organizational factors influencing nursing care rationing in ICUs, providing valuable insights for improving nursing care quality and patient outcomes in these critical settings.

6. PLOS authors have the option to publish the peer review history of their article (what does this mean?). If published, this will include your full peer review and any attached files.

Reviewer #1: No

Reviewer #2: **Yes: **Izabella Uchmanowicz

---

## [Author Response · Author response to Decision Letter 0]

27 Mar 2024

We want to express our appreciation to the reviewers and the editorial board for taking the time and effort necessary to review our manuscript in Plos One. We carefully considered your comments and introduced changes in the manuscript accordingly to reviewers recommendations. The manuscript has been completely rewritten. All changes are listed in a separate file. Thank you for your consideration. I look forward to hearing from you.

Sincerely,

Anna Antoszewska, MSc, RN

---

## [Decision Letter · Decision Letter 1]

4 Jun 2024

PONE-D-24-02427R1Rationing nursing care and organizational factors in intensive care unitsPLOS ONE

Dear Dr. Antoszewska,

Thank you for submitting your manuscript to PLOS ONE. After careful consideration, we feel that it has merit but does not fully meet PLOS ONE’s publication criteria as it currently stands. Therefore, we invite you to submit a revised version of the manuscript that addresses the points raised during the review process. **One of the original reviewers has provided some further recommendations that should be addressed in another revision of the manuscript.**

Kind regards,

Sascha Köpke

Academic Editor

PLOS ONE

Journal Requirements:

Reviewers' comments:

Reviewer's Responses to Questions

**Comments to the Author**

1. If the authors have adequately addressed your comments raised in a previous round of review and you feel that this manuscript is now acceptable for publication, you may indicate that here to bypass the “Comments to the Author” section, enter your conflict of interest statement in the “Confidential to Editor” section, and submit your "Accept" recommendation.

Reviewer #1: All comments have been addressed

2. Is the manuscript technically sound, and do the data support the conclusions?

Reviewer #1: Yes

3. Has the statistical analysis been performed appropriately and rigorously? 

Reviewer #1: Yes

4. Have the authors made all data underlying the findings in their manuscript fully available?

Reviewer #1: Yes

5. Is the manuscript presented in an intelligible fashion and written in standard English?

Reviewer #1: Yes

6. Review Comments to the Author

**Reviewer #1:** Dear authors

Thank you for addressing my comments. I have some comments as follows:

• Page 2, line 19: please change “Omitted nursing care” to “rationing of nursing care”

• Page 2, lines 21-22: Please add the abbreviation “intensive care units” in parentheses as follows: Intensive Care Units (ICUs).

• Page 2, line 40: please use the “ICUs” instead of the “intensive care units”.

• Page 2, line 41: add the term of “organizational factors” to keywords.

• Page 3, lines 44-45: I suggest removing the abbreviations MCN and MNCM because they are used only once in the entire manuscript.

• Page 7, line 147: Please write "neonatal ICUs" instead of "ICUs for neonates".

• I suggest that the text of lines 156 to 162 be moved to the ethical considerations section and merged with it so that sentences do not become repetitive. Because some sentences overlap.

• Table 1: Please mention the age unit in the related column in the table in parentheses as follows: age (years)

• Table 1, the column related to "the number of hours spent at work per month": please put the word "hours" in parentheses in the first column and remove it from the rest of the columns as follows: number of hours spent at work per month (hours): write "100-180" instead of "100-180 hours" ,....

• Table 3: Please mention the age unit in the related column in the table in parentheses as follows: age (years)

Best regard

7. PLOS authors have the option to publish the peer review history of their article (what does this mean?). If published, this will include your full peer review and any attached files.

Reviewer #1: No

---

## [Author Response · Author response to Decision Letter 1]

6 Jun 2024

We thank the reviewers and the editorial board for taking the time and effort necessary to review our manuscript in Plos One, titled “Rationing nursing care and organizational factors in Intensive Care Units.” We carefully considered your comments and introduced changes in the manuscript according to reviewers' recommendations. All changes are listed in a separate file.

---

## [Editor Report · Decision Letter 2]

17 Jun 2024

Rationing nursing care and organizational factors in intensive care units

PONE-D-24-02427R2

Dear Dr. Antoszewska,

We’re pleased to inform you that your manuscript has been judged scientifically suitable for publication and will be formally accepted for publication once it meets all outstanding technical requirements.

Kind regards,

Sascha Köpke

Academic Editor

PLOS ONE
---

## [Editor Report · Acceptance letter]

19 Jun 2024

PONE-D-24-02427R2 

PLOS ONE

Dear Dr. Antoszewska, 

I'm pleased to inform you that your manuscript has been deemed suitable for publication in PLOS ONE. Congratulations! Your manuscript is now being handed over to our production team.

Kind regards, 

on behalf of

Professor Sascha Köpke 

Academic Editor

PLOS ONE